# GA-Dueling DQN Jamming Decision-Making Method for Intra-Pulse Frequency Agile Radar

**DOI:** 10.3390/s24041325

**Published:** 2024-02-19

**Authors:** Liqun Xia, Lulu Wang, Zhidong Xie, Xin Gao

**Affiliations:** 1National Innovation Institute of Defense Technology, Academy of Military Science, Beijing 100010, China; xialiqun2022@163.com; 2Intelligent Game and Decision Laboratory, Academy of Military Science, Beijing 100091, China; 3Chinese People’s Liberation Army 32806 Unit, Academy of Military Science, Beijing 100091, China; 4The 85th Detachment, Chinese People’s Liberation Army 95969 Unit, Wuhan 430000, China; gaoxin072926@163.com

**Keywords:** optimizing jamming strategies, jamming-to-noise ratio, deep reinforcement learning

## Abstract

Optimizing jamming strategies is crucial for enhancing the performance of cognitive jamming systems in dynamic electromagnetic environments. The emergence of frequency-agile radars, capable of changing the carrier frequency within or between pulses, poses significant challenges for the jammer to make intelligent decisions and adapt to the dynamic environment. This paper focuses on researching intelligent jamming decision-making algorithms for Intra-Pulse Frequency Agile Radar using deep reinforcement learning. Intra-Pulse Frequency Agile Radar achieves frequency agility at the sub-pulse level, creating a significant frequency agility space. This presents challenges for traditional jamming decision-making methods to rapidly learn its changing patterns through interactions. By employing Gated Recurrent Units (GRU) to capture long-term dependencies in sequence data, together with the attention mechanism, this paper proposes a GA-Dueling DQN (GRU-Attention-based Dueling Deep Q Network) method for jamming frequency selection. Simulation results indicate that the proposed method outperforms traditional Q-learning, DQN, and Dueling DQN methods in terms of jamming effectiveness. It exhibits the fastest convergence speed and reduced reliance on prior knowledge, highlighting its significant advantages in jamming the subpulse-level frequency-agile radar.

## 1. Introduction

Electronic Warfare (EW) is a critical component of modern warfare. It enables utilization of the electronic spectrum and electromagnetic waves to interfere with, deceive, or disable the attacking electronic systems. With the rapid development of artificial intelligence in recent years, Cognitive Electronic Warfare (CEW) has emerged as a new focus. CEW can be characterized as an intelligent EW system that autonomously detects the electromagnetic environment. CEW dynamically adjusts its jamming strategies in real time through learning and reasoning. It also evaluates the impact of its actions to effectively counter threats [1].

In contrast to traditional electronic warfare, which depends on pre-designed jamming signals, Cognitive Electronic Warfare (CEW) detects spectral shifts and autonomously adjusts jamming signal parameters and characteristics to optimize jamming effectiveness Intelligent jammers capable of autonomous learning and adapting to dynamic environments referred to as cognitive jammers. The Frequency Agile Radar (FA Radar) reduces the effectiveness of jamming by employing proactive frequency hopping [2]. The Pulse Frequency Agile Radar alters frequencies between pulses, while the Intra-Pulse Frequency Agile Radar divides a single pulse into multiple sub-pulses, enabling frequency variation within the pulse. This significantly complicates predictive jamming efforts to discern the hopping patterns.

Reinforcement learning is a machine learning approach where agents learn optimal actions through continuous interaction, trial and error, and feedback from the environment [3]. Reinforcement learning provides the advantages of online interactive learning and strategy updating without requiring prior information. Deep Reinforcement Learning, which combines deep learning with reinforcement learning, can acquire more sophisticated strategies. Due to the powerful fitting capabilities of deep neural networks, Deep Reinforcement Learning can automatically extract high-dimensional features from raw states, effectively handling complex, high-dimensional, and nonlinear state spaces. Employing reinforcement learning for jamming strategy optimization is a key method in addressing intelligent jamming decisions in complex and dynamically changing electromagnetic environments.

Research in this field can be categorized into two types. The first type centers on radar jamming decisions using reinforcement learning, with a focus on selecting appropriate jamming patterns. These methods aim to optimize the jamming effects for multifunctional radars by choosing suitable patterns instead of adjusting specific jamming parameters. The second type of research focuses on the selection of specific jamming parameters, mainly in the context of communication jamming. These studies focus on the impact of jamming signals on communication systems, with the objective of disrupting the signal and consequently degrading or severing communication links by adjusting jamming parameters.

Optimization of radar jamming strategies is a pivotal step in cognitive electronic warfare, particularly in the selection of jamming patterns or parameter optimization to achieve tactical objectives in radar countermeasures. As radar systems become increasingly intelligent and adaptive, the effectiveness and timeliness of traditional template-matching jamming decision methods are facing challenges.

In recent years, reinforcement learning has been widely applied in communication, anti-jamming, and jamming strategies [4,5,6,7,8,9], bringing new ideas and methods to the field of radar jamming. Xing Q et al. [10] studied multiple operational modes of a multifunction phased array radar and examined the jamming decision-making problem using the Q-learning method, where changes in radar operational modes were utilized as jamming rewards. Zhang B et al. [11] proposed a jamming decision-making approach based on deep Q neural networks and analyzed the influence of prior knowledge on jamming decision-making. In [12], the authors modeled the transitions of radar operational states and employed reinforcement learning to select appropriate jamming modes. In [13], the authors also focused primarily on jamming strategies for the macroscopic operational modes of the radar. Zhang et al. [14] proposed a collaborative jamming decision method based on reinforcement learning. By introducing coordinated jamming strategies and establishing scenarios of multiple jammers countering multi-functional networked radars, researchers proposed a dual-deep Q-network based on prioritized experience replay. The collection of jamming strategies in the aforementioned works only includes different jamming patterns, which belong to coarse granularity and low degrees of freedom in jamming decision making. They do not fully consider the flexible actions taken by radar in each mode, such as frequency agility and dynamic pulse repetition interval. These agile actions within the modes have a significant impact on jamming effectiveness.

In addition, some scholars have conducted more in-depth research on intelligent jamming decision-making problems. They considered not only jamming patterns but also various jamming strategies, including jamming modulation parameters and power, to achieve intelligent optimization of jamming parameters for cognitive jamming devices. In [15], the authors used Q-learning to solve for the optimal jamming frequency of the jamming device. They iteratively updated and learned the best jamming frequency through reinforcement learning, resulting in a significant improvement in the jamming-to-noise ratio. A two-level jamming decision-making framework was proposed in [16], which utilizes dual-layer Q-learning and dynamic jamming evaluation methods to optimize the jamming strategy. By decomposing the high-dimensional jamming space into two low-dimensional subspaces of jamming patterns and pulse parameters, the framework achieves the selection of the optimal solution in the two-level Q-learning and accelerates the convergence rate. Additionally, in [17], the authors investigated the intelligent jamming device’s selection and execution of jamming tasks based on jamming type and power in radar adversarial scenarios and presented a joint optimization method based on deep reinforcement learning. In [18], the authors modeled the process of jammers penetrating netted radar systems as a Markov Decision Process. They utilized a proximal policy optimization algorithm to jointly learn the optimal path and jamming power allocation strategy. The learning process was guided by a reward function that depended on the success or failure of penetration and the distance to the target. In [19], the competition between frequency-agile radar and jamming devices under incomplete information is investigated, and a deep learning approach is employed to solve the game problem. Furthermore, reinforcement learning has numerous applications in communication adversarial scenarios. In [20,21,22], the authors extensively studied the utilization of reinforcement learning algorithms to search for optimal jamming strategies at the physical layer and the MAC layer.

However, the ability of frequency-agile radars to rapidly change carrier frequencies or switch them completely at random poses a challenge for traditional jamming decision-making methods, making it difficult to address this situation. Currently, research on intelligent jamming strategies specifically targeting this scenario is relatively limited.

Existing literature has examined the optimization of jamming patterns for multifunctional radars using reinforcement learning, as well as the optimization of jamming parameters, including jamming power and jamming frequency, based on radar waveform parameters. This paper focuses on the optimization of jamming parameters for frequency-agile radars. For intra-pulse frequency-agile radars, which exhibit a significant frequency change space, a deep reinforcement learning algorithm is considered for jamming frequency selection. Using interactive learning, the agile jamming strategy for intra-pulse frequency-agile radars is acquired, thereby enhancing the effectiveness of targeted jamming.

The main contributions of this paper are summarized as follows:We construct models of an intra-pulse frequency-agile radar and a cognitive jammer and formulated scenarios for simulating adversarial jamming situations. To fully utilize the electronic warfare capabilities of frequency-agile radars, the radar model is designed with the capability to randomly select frequency bands both between and within pulses. The support jammer is equipped with the ability to perceive environmental changes and adaptively adjust its jamming strategy in response. The intelligent jamming strategy is abstracted as an optimization problem of selecting the optimal jamming frequency.We model the interaction between the jammer and the radar as a Markov Decision Process (MDP). We propose a jamming frequency selection method based on GA-Dueling DQN (GRU-Attention based Dueling Deep Q Network), which leverages the long-term dependency capturing ability of Gated Recurrent Units (GRU) and incorporates attention mechanisms. By learning the radar’s frequency variation patterns during the interaction, the jam-to-signal ratio is enhanced, the targets are protected, and the probability of being detected by the radar is reduced.Extensive experimental comparisons are conducted to validate the effectiveness and superiority of the proposed algorithm in this paper. Compared to traditional reinforcement learning decision methods such as Q-learning, DQN, and Dueling DQN, the proposed method in this paper exhibits better jamming performance, faster convergence speed, and lower reliance on prior knowledge. Specifically for the fully randomly switched carrier frequency sub-pulse frequency-agile radar, the method proposed in this paper successfully jams each sub-pulse with a probability of 97.14%, demonstrating significant advantages.

The organizational structure of this paper is as follows: Section 2 introduces the model of intra-pulse frequency agile radar and the scenario of jamming countermeasures. Section 3 formulates the cognitive jamming decision model as a Markov decision process and applies deep reinforcement learning algorithms for intelligent jamming decision making. Section 4 presents comprehensive simulation analysis evaluating the effectiveness of the proposed intelligent jamming decision method. Section 5 provides a summary of the paper and offers prospects for future work.

## 2. The Analysis of Frequency Agile Radar and Cognitive Jammer Models

### 2.1. Model of Intra-Pulse Frequency Agile radar

A frequency-agile radar is a radar system capable of randomly switching carrier frequencies between pulses, thereby demonstrating excellent anti-jamming capabilities. In [23], a specific type of frequency agile radar called intra-pulse frequency agile radar is introduced. In this radar system, a single pulse consists of multiple sub-pulses, and carrier frequency can be switched between these sub-pulses. This capability further enhances the radar’s electronic countermeasure capabilities.

As shown in Figure 1, the sequence of pulses transmitted by the radar is illustrated. The sequence includes n=1,2,⋯,N, n pulses in total, with a fixed pulse repetition period denoted as *T*. Each pulse is divided into *K* sub-pulses, where $f_{k}^{\left(n\right)}$ represents the carrier frequency used for the kth sub-pulse of the *n*th pulse.

Therefore, the transmission signal of a radar pulse can be represented as
(1)$$s_{T}^{\left(n\right)}\left(t\right)=\sum_{k=0}^{K-1}\mathrm{rect}\left(t-kT_{c}\right)u\left(t-kT_{c}\right)\exp\left(j2\pi f_{k}t\right),$$
where u(t) represents the complex envelope of the transmitted signal. *K* represents the number of sub-pulses in a pulse. $T_{c}$ denotes the time duration of each sub-pulse. $T_{p}$ is cumulative time duration of the *K* sub-pulses. $f_{k}$ represents the carrier frequency of the k-th sub-pulse. rect(t) represents the rectangular function defined as
(2)$$\mathrm{rect}\left(t\right)=\left\{\begin{array}{cc} 1 & 0⩽t⩽T_{c}\\ 0 & \;\mathrm{otherwise}\; \end{array}\right..$$

Assuming the radar’s frequency band is defined as $F=[f_{L},f_{H}]$, with a bandwidth of $B=f_{H}-f_{L}$, bandwidth *B* is divided into *M* equally spaced subbands. Each subband has a range of $F=f_{L}+d_{n}\Delta f,\;f_{L}+\left(d_{n}+1\right)\Delta f$, where $d_{n}$ is the frequency modulation code, a random integer less than B/Δf−1, and the frequency change step of Δf=B/M. Within a sequence of pulses, the radar can randomly select different subbands for each pulse. The carrier frequency of consecutive pulses in the radar pulse train jumps with an interval of Δf. Additionally, the frequency-agile radar also has the capability of frequency agility at the sub-pulse level. Each pulse contains *K* sub-pulses with different frequencies. All *K* sub-pulses collectively occupy the subband of the corresponding pulse, with a bandwidth of Δf.

As an example, Figure 2 illustrates the possible frequency selection scheme for the sub-pulses.

### 2.2. Radar Countermeasure Scenarios

In the depicted scenario shown in Figure 3, there is a frequency-agile radar, a target, and a supporting cognitive jammer. The frequency-agile radar, as described previously, possesses sub-pulse-level frequency agility and can randomly select frequency bands between pulses. The supporting jammer is capable of sensing environmental changes and adapting its jamming strategy accordingly, with a primary focus on jamming frequencies. The mission of the jammer is to protect the target by reducing the probability of detection by the radar. Therefore, the Jamming-to-Signal ratio (JSR) is chosen as the objective function for the jammer.

We assume that the target is a point target with an RCS of σ. The transmit powers of the radar and jammer, denoted as $P_{R}$ and $P_{J}$, respectively, depend on their pulse amplitudes and pulse widths. The radar pulse width is denoted as $\tau^{\left(R\right)}$, the pulse amplitude is $A_{R}$, and the pulse repetition period is *T*. The radar transmit power is given by
(3)$$P_{R}=\frac{A_{R}^{2}\tau^{\left(R\right)}}{2T}.$$

We assume that the cognitive jammer also possesses frequency agility at the sub-pulse level and can select frequencies within the frequency band range of the frequency-agile radar. The cognitive jammer has access to the radar’s entire frequency band and Pulse Repetition Interval (PRI) information through Electronic Intelligence (ELINT) systems. Therefore, the frequency strategy selection space of the cognitive jamming aircraft aligns completely with the frequency variation space of the frequency-agile radar.

The jamming aircraft has a pulse width of $\tau^{\left(J\right)}$ and a pulse amplitude of $A_{J}$, with the same pulse repetition period, *T*. Therefore, the jamming aircraft’s transmit power can be calculated as follows:(4)$$P_{J}=\frac{A_{J}^{2}\tau^{\left(J\right)}}{2T}.$$

The received Jamming-to-Signal ratio at the radar can be expressed as the product of the target-to-radar channel gain $h_{t}$, the jammer-to-radar channel gain $h_{j}$, and the assumed sidelobe loss *L*. Therefore, the radar’s received jam-to-signal ratio can be represented as
(5)$$JSR=\frac{P_{J}h_{j}L}{P_{R}h_{t}^{2}\sigma }.$$

Assuming the cognitive jammer is capable of environmental sensing and can estimate parameters, such as carrier frequency information from the received radar signals, it can determine whether the jamming frequency aligns with radar frequency at the next moment. By maximizing the Jamming-to-Signal ratio, as described earlier, the cognitive jammer can reduce the attacker’s radar’s probability of detecting the target, thus protecting the target from being detected.

The cognitive jammer aims to decrease the signal-to-noise ratio of the radar’s received signal by maximizing the Jamming-to-Signal ratio, thereby impeding the radar’s ability to detect and accurately track the target.

## 3. Intelligent Jamming Decision Methods Based on Deep Reinforcement Learning

In this section, we start by reviewing the concepts and definitions of reinforcement learning and model the interaction between the jammer and the radar as a Markov Decision Process (MDP). Finally, we present a detailed description of the proposed intelligent jamming decision algorithm.

### 3.1. Introduction to Reinforcement Learning

Reinforcement Learning (RL) is a machine learning approach that involves the interaction between an agent and an environment. The agent learns through trial and error to maximize cumulative rewards and achieve task optimization.

The Markov Decision Process (MDP) is a fundamental concept in RL and serves as a mathematical model for describing sequential decision problems. In an MDP, the agent interacts with the environment by making decisions based on observed states, taking actions, receiving rewards, and transitioning to new states. The immediate rewards obtained depend on both the current state and the chosen actions. This iterative process continues until the task is completed. A Markov Decision Process can be denoted as <𝒮,𝒜,𝒫,ℛ>.

An MDP can be described using the following elements:𝒮: The set of environment states that the agent can observe.𝒜: The set of actions that the agent can take in each state.𝒫: The transition probability, $P_{a}\left(s,s^{\prime }\right)=P\left(s_{t+1}=s^{\prime }|s_{t}=s,a_{t}=a\right)$, which represents the probability of transitioning from state *s* to state $s^{\prime }$ when taking action *a* in state *s*.ℛ: The reward function, $R_{s}^{a}=E\left[r_{t+1}|s_{t}=s,a_{t}=a\right]$, which represents the expected immediate reward that the agent can receive when taking action *a* in state *s*.

In an MDP, the agent’s objective is to maximize the long-term cumulative reward, also known as the expected return, $U_{t}$,
(6)$$U_{t}=R_{t}+\gamma \cdot \sum_{k=t+1}^{n}\gamma^{k-t-1}\cdot R_{k},$$
where γ is the discount factor, which measures the importance of future rewards, and $R_{t}$ is the reward generated from the tth interaction. In order to maximize the long-term cumulative reward, the agent needs to learn policy π(a|s), which is the probability distribution of selecting action *a* in state *s*, to maximize the expected return. Value function $V_{\pi }\left(s\right)$ is used to evaluate learned policy π(a|s). The value function can be categorized into state value function and action value function, represented by Equations (7) and (8), respectively. The action value function is also known as the Q-value function.
(7)$$V_{\pi }\left(s\right)=E_{\pi }\left[U_{t}|S_{t}=s\right],$$
(8)$$Q_{\pi }\left(s,a\right)=E_{\pi }\left[U_{t}|S_{t}=s,A_{t}=a\right].$$

Q-learning is a reinforcement learning algorithm that learns state-action value function $Q_{\pi }\left(s,a\right)$ to achieve optimal policy selection for an agent without knowledge of the environment model. The Q-value function represents the expected cumulative reward obtained by taking different actions in the current state. The core idea of the Q-learning algorithm is to iteratively update the Q-value function, enabling the agent to learn the optimal policy through continuous interactions with the environment.

The Q-learning algorithm iteratively updates the Q-value function using Equation (Equation 9), enabling the agent to learn the optimal action-value function $Q^{*}\left(s_{t},a_{t}\right)=\max_{\pi }Q_{\pi }\left(s_{t},a_{t}\right)$.
(9)$$\begin{array}{c} Q^{*}\left(s_{t},a_{t}\right)=E_{S_{t+1}∼p\left(\cdot |s_{t},a_{t}\right)}\left[R_{t}+\gamma \cdot \max_{A\in 𝒜}Q^{*}\left(S_{t+1},A\right)|S_{t}=s_{t},A_{t}=a_{t}\right] \end{array}.$$

Based on Monte Carlo approximation, we can obtain the following:(10)$$Q^{*}\left(s_{t},a_{t}\right)\approx r_{t}+\gamma \cdot \max_{a\in 𝒜}Q^{*}\left(s_{t+1},a\right).$$

The Q-learning algorithm uses a table to store the Q-values for each state-action pair. However, when confronted with high-dimensional state spaces, Q-learning faces challenges in learning the optimal policy function.

DQN (Deep Q-Network) is a deep reinforcement learning algorithm based on Q-learning. Its core idea is to approximate $Q^{*}\left(s_{t},a_{t}\right)$ using a deep neural network, denoted as $Q\left(s_{t},a_{t};w\right)$, where w represents the parameters of the neural network. By leveraging the powerful approximation capabilities of neural networks, DQN overcomes the limitations of traditional Q-learning algorithms in high-dimensional state spaces and continuous action spaces. It also employs techniques such as Experience Replay and Target Network to stabilize the training process. In Experience Replay, the agent stores the experienced state-action-reward sequences in a replay buffer and randomly samples from it for training. The iterative update of the Q-value function in DQN is shown in Equation (Equation 11),
(11)$$\underset{\;\hat{q_{t}}}{\underset{︸}{Q\left(s_{t},a_{t};w\right)}}\approx \underset{\mathrm{TD}\;\mathrm{target}\;\hat{y_{t}}}{\underset{︸}{r_{t}+\gamma \cdot \max_{a\in 𝒜}Q\left(s_{t+1},a;w\right)}}.$$

$Q\left(s_{t},a_{t};w\right)$ represents the prediction made by the neural network at time step *t*, and $r_{t}+\gamma \cdot \max_{a\in 𝒜}\left(s_{t+1},a;w\right)$ represents the prediction made by the neural network at time step t+1, also known as the TD target $\hat{y_{t}}$, where $r_{t}$ is the observed reward. The part of $\hat{y_{t}}$ that is based on facts is more reliable than $\hat{q_{t}}$, so $\hat{q_{t}}$ should be made to approximate $\hat{y_{t}}$, and the difference between them is called the TD error. The loss function is defined as shown in Equation (Equation 12), and the gradient of L(w) with respect to parameters w is computed to update the parameters, where α in Equation (Equation 14) denotes the learning rate:(12)$$L\left(w\right)=\frac{1}{2}{\left[Q\left(s_{t},a_{t};w\right)-{\hat{y}}_{t}\right]}^{2},$$
(13)$$\nabla_{w}L\left(w\right)≜\left({\hat{q}}_{t}-{\hat{y}}_{t}\right)\cdot \frac{\partial Q(s,a;w)}{\partial w},$$
(14)$$w\leftarrow w-\alpha \cdot \left({\hat{q}}_{t}-{\hat{y}}_{t}\right)\cdot \frac{\partial Q(s,a;w)}{\partial w}.$$

Q-learning and DQN are reinforcement learning algorithms based on value function estimation which use value functions to evaluate the merits of policies. Through interactive learning between the agent and the environment, the optimal action value function $Q_{\pi }^{*}\left(s,a\right)$ can be approximated so that $\pi \left(s\right)=\underset{a}{\mathrm{argmax}}Q^{*}\left(s,a\right),\forall s\in S$, thereby obtaining the optimal policy. Below, the interaction between the cognitive jammer and the intra-pulse frequency agile radar is modeled as a Markov decision process (MDP), and reinforcement learning algorithms are utilized to solve the frequency optimization selection problem in the intelligent jamming decision of the jammer.

### 3.2. Intelligent Jamming Decision Model

In this section, the interaction between the cognitive jammer and the intra-pulse frequency agile radar is modeled as a Markov decision process, providing a basis for constructing a model for jammer intelligent decision-making using deep reinforcement learning.

The cognitive jammer can be viewed as an agent, while the signal transmitted by the frequency agile radar serves as dynamically changing environmental information. Through interaction with the frequency agile radar, the cognitive jammer can use the Jamming-to-Signal ratio as a jamming reward function to record the jamming effect of each pulse. The optimization objective of the jammer strategy is to maximize the total Jamming-to-Signal ratio within the pulse train as a reward.

Figure 4 illustrates the interaction framework between the intra-pulse frequency agile radar and the cognitive jammer based on reinforcement learning.

Based on the modeling of the radar and the jammer, the state space is the set of all frequencies of the frequency agile radar, i.e., $S_{t}=f_{t}^{\left(r\right)}$, where $f_{t}^{\left(r\right)}\in ℱ$, $f_{t}^{\left(r\right)}$ is the frequency of the pulse signal transmitted by the radar. According to the frequency agile radar model described in the previous section, the number of states contained in this state space is K×M. It is assumed that the jammer’s action space is identical to the state space, i.e., the action space of the cognitive jammer is $A_{t}=f_{t}^{\left(j\right)}$, where $f_{t}^{\left(j\right)}$ is the frequency of the pulse signal transmitted by the cognitive jammer. The total number of actions is K×M, which is consistent with the total number of states. Reward function $r_{t}$ for each interaction is the Jamming-to-Signal ratio (JSR) of the radar receiver. According to Equation (Equation 15), it is calculated as
(15)$$\begin{array}{cc} r_{t} & =JSR_{t}\\ \; & =\frac{P_{J}h_{j}L}{P_{R}h_{t}^{2}\sigma }*Num\\ \; & =\frac{A_{J}^{2}\tau_{t}^{\left(J\right)}h_{j}L}{A_{R}^{2}\tau_{t}^{\left(R\right)}h_{t}^{2}\sigma }*Num. \end{array}$$

Here, Num is the number of radar subpulses that are equal to the jamming subpulses. The goal of the cognitive jammer is to find the jamming frequency selection policy, $\pi^{*}\left(a|s\right)$, that has the maximum average Jamming-to-Signal ratio.

At time *t*, the jammer is able to capture radar pulse signal $s_{t}$ at time *t*, and takes the optimal jamming action, $a_{t}$, according to policy function π. After the jammer executes the action, the environment state transitions from $s_{t}$ to $s_{t+1}$. The reward, $r_{t}$, is the Jamming-to-Signal ratio received by the radar.

In the jammer’s decision process, the jamming effect of each pulse can be measured by the Jamming-to-Signal ratio. By choosing appropriate jammer policies and parameters, the jammer can optimize the total Jamming-to-Signal ratio within the pulse train. The cumulative expected return is the sum of all Jamming-to-Signal ratio obtained by the jammer within a pulse repetition period. For the jammer, its goal is to maximize the total Jamming-to-Signal ratio within the pulse repetition period.

We suppose the jammer has the ability to observe the environment through the intercept receiver, allowing for it to analyze whether the radar will be jammed in the upcoming pulse. The reward obtained in the next pulse serves as feedback to the jammer. Additionally, assuming the jammer is equipped with the capability to sense the radar signal and estimate parameters such as carrier frequency and pulse width, it can make estimations in the next time step regarding the jamming status of the radar, as well as estimate the Jamming-to-Signal Ratio.

### 3.3. Deep Reinforcement Learning-Based Intelligent Jamming Decision Approach

This paper proposes an improved algorithm based on Dueling DQN [24] to solve the intelligent jamming decision problem modeled as an MDP. In this algorithm model, the cognitive jammer’s goal is to find the optimal policy, π, that has the maximum average reward. During the learning phase, the jammer samples a small batch of experiences from the prioritized experience replay buffer (PER) according to non-uniform weights to update the network.

Q-learning is a classic method for solving the Markov decision model described in the previous section. However, due to its limited fitting capability, DQN methods are usually used to solve complex models with high-dimensional state spaces by leveraging the powerful fitting capability of deep neural networks. In Equation (Equation 9), Q-learning uses its own estimates to update Q-values, which leads to bias propagation. Moreover, the maximization operation often results in overestimating the TD targets, causing the DQN trained with Q-learning to suffer from an overestimation problem, particularly in a non-uniform manner. This drawback negatively impacts the performance of DQN. To alleviate overestimation, as shown in Figure 5, this paper adopts a double Q-learning algorithm [25] to update parameters. The first step involves selecting an action that maximizes the output of the DQN, based on state *s*, using the policy network. The second step calculates the value using the target network. By alternating between these two networks, it is possible to improve both the accumulation of biases and the overestimation resulting from maximization simultaneously. Additionally, in traditional Q-learning, updating the value function involves using the maximum action value of the current state. However, this approach may lead to increased volatility in the value function due to the potential instability of selecting the maximum action value. Double Q-learning mitigates this volatility and enhances learning stability by assigning the maximization operation to two independent Q-functions.

The Dueling DQN decomposes the action value function, Q(s,a), into a state value function, V(s), and an advantage function, A(s,a). V(s) predicts the expected return of a state, while A(s,a) calculates the relative advantage of each action compared to the expected return. Finally, V(s) and A(s,a) are combined to calculate the Q value for each action, as shown in the network structure in Figure 6 (left).This decomposition enables the cognitive jammer to learn the relative advantage of each action instead of independently learning the value of each action. Even when the value differences between certain actions in specific states are small, the cognitive jammer can still accurately assess the merit of each action. Furthermore, by centralizing the advantage function, the overestimation problem in traditional DQN can be alleviated and the stability of learning can be improved. The specific way of centralization is to subtract the mean of the advantage function A(s,a). With this operation, the learning target of the advantage function A(s,a) network is closer to the state value function rather than being dominated by the overestimated action advantages.
(16)$$V^{*}\left(s\right)=\max_{\pi }V_{\pi }\left(s\right),$$
(17)$$A^{*}\left(s,a\right)≜Q^{*}\left(s,a\right)-V^{*}\left(s\right),$$
(18)$$\begin{array}{cc} & Q\left(s,a;w\right)≜V\left(s;w^{V}\right)+A\left(s,a;w^{A}\right)-\max_{a\in A}A\left(s,a;w^{A}\right) \end{array}.$$

In the algorithm framework shown in Figure 5, both the policy network and the target network adopt the improved dueling network designed in this paper: GRU-Attention-based Dueling DQN (GA-Dueling DQN), whose network structure is shown in Figure 6 (right).

The GA-Dueling DQN method proposed in this paper designs an NN network module compared to the original Dueling DQN method, whose structure is shown in Figure 7.

Even though Dueling DQN can effectively alleviate the overestimation problem and improve the stability of learning by combining techniques like double Q-learning and prioritized experience replay buffer, it has difficulty providing sufficient information from single observations due to the random generation of radar frequencies. The GRU layer is a variant of a recurrent neural network (RNN) that can remember and update past state information through a self-feedback mechanism [26]. By using the GRU layer, the model can capture long-term dependency relationships in the sequence by utilizing historical observations, thereby better understanding the pattern of changes in radar frequencies. Consequently, at each time step, the observed environment state, i.e., the frequency of the pulse signal transmitted by the radar, is first input to the GRU layer, enabling the cognitive jammer to switch the frequency of the jamming pulse signal by combining current and historical observations, which solves the problem of insufficient single observations for the cognitive jammer. Next, an eight-headed multi-head self-attention module is employed. This module leverages the attention mechanism to learn correlations between different input features, resulting in improved extraction of relevant features and representations from input observations [27]. The output of the multi-head attention module is then processed by a linear layer, which further transforms it into feature representations with higher dimensions and richer semantic information. Lastly, the processed features are fed into two identical linear layers, serving as inputs to the value network and the advantage network. This design facilitates more effective learning of key information from the input observation sequence.

Furthermore, the Noisy Net [28] technique is employed in the linear layers, $FC_{i}$, by way of introducing Gaussian noise to the weights and biases of the neural network. The addition of layer normalization after the linear layer helps alleviate the issues of gradient vanishing and exploding, improves the convergence of the network, and enhances its generalization capability. This augmentation promotes exploration and enhances learning. Figure 8 illustrates the structures of state value network V(s), advantage network A(s,a) and linear layers $FC_{i}$. Finally, the outputs of these two networks are calculated through Equation (18) to obtain Q(s,a), the *Q* values of all jamming pulse frequencies that can be taken by the jammer.

In summary, the proposed model utilizes a GRU layer to leverage historical observations and capture the long-term dependencies in radar frequency changes. Additionally, it employs a multi-head self-attention mechanism to learn correlations between different features and extract relevant representations. The processed features are then fed into linear layers as inputs to the value network and advantage network in Dueling DQN. Furthermore, the model enhances its exploratory and reinforcement learning capabilities by incorporating the Noisy Net technique.

At each time step, the jammer selects the frequency using an ε-greedy algorithm based on the current state’s Q-values. It selects the action with the largest Q-value, with a probability of 1 − ε, while randomly choosing actions with a probability of ε. This approach is commonly used to balance exploration and exploitation. Exploration rate ε follows an exponential decay schedule from 0.995 to 0.005, maintaining a consistent decay rate.

Algorithm 1 follows the following procedure:
**Algorithm 1** Intelligent Jamming Decision-Making Algorithm Based on GA-Dueling DQN1:**Input:** Radar pulse signal frequency: $f_{t}^{\left(r\right)}$2:**Output:** Jammer pulse signal frequency: $f_{t}^{\left(j\right)}$3:**Initialize:** weights of $Q_{\mathrm{state}},Q_{\mathrm{advantage}}$, and $Q_{\mathrm{value}}$ networks, learning rate α, discount factor γ, exploration rate ε, and prioritized experience replay buffer *B*4:**for** each $s_{t}=f_{t}^{\left(r\right)}$∈ radar pulse train **do**5:  Input state $s_{t}$ into GA-Dueling DQN module,6:  Calculate the Q-values for each action according to Equation (Equation 18),7:  Select action $a_{t}$ according to the ε-greedy strategy,8:  Obtain $s_{t+1}$ and calculate the $r_{t}$,9:  Add transition $\left(s_{t},a_{t},r_{t+1},s_{t+1}\right)$ to replay buffer *B*.10:**end for**11:Sample a Minibatch of transitions from the replay buffer *B* based on priorities12:**for** each transition $\left(s,a,r,s^{\prime }\right)$
**do**13:  Calculate TD target $y_{t}$ according to Equation (Equation 11),14:  Calculate the current state-value according to Equation (Equation 17),15:  Update the $Q_{\mathrm{state}},Q_{\mathrm{advantage}}$, and $Q_{\mathrm{value}}$ networks according to Equation (Equation 14).16:**end for**

## 4. Simulation Analysis

In this section, simulation experiments are conducted to verify the effectiveness of the proposed method and compare it with traditional Q-learning, DQN and original Dueling DQN methods. In the simulation, it is assumed that a radar pulse train contains 10,000 pulses. The carrier frequency range of the radar is *F* = [10 GHz, 11 GHz], where B=1 GHz. The frequency band is equally divided into N=10 subbands with a 100 MHz interval as the frequency range of pulses. Subpulse width $T_{c}$ is assumed to be 40 ns. Each pulse has K=4 subpulses, so the subpulse bandwidth is 25 MHz. To fully utilize the bandwidth, the frequencies of the four subpulses are pairwise distinct, so there are 24 combinations in total. That is, the state space of the frequency agile radar is 24×10=240. The action space of the jammer is the same as the state space.

The parameter settings of the radar, the jammer and the network are shown in Table 1.

### 4.1. Frequency Selection by Jammer

The GA-Dueling DQN method proposed in this paper is compared with original Dueling DQN, DQN and Q-learning methods. Table 2 presents the parameters of the improved method, as well as the network structures of DQN and Dueling DQN. The only difference between DQN, Dueling DQN and the improved method is that the GRU layer and multi-head attention layer are replaced with linear layers. Other parameters such as exploration rate, discount factor, etc., are set exactly the same for the three methods. Both DQN and Dueling DQN adopt cosine learning rate warm-up techniques with a maximum learning rate of 0.009.

Employing the aforementioned three methods, one radar pulse train is taken as one round of interaction process, and each round of confrontation process contains 10,000 pulses as a pulse train. $1\times 10^{6}$ The experiment conducts 100 rounds of confrontation, and the total reward of the jammer in each round of confrontation process is compared as shown in Figure 9.

Figure 9 illustrates that, for Q learning, DQN, Dueling DQN, and the GA-Dueling DQN proposed in this paper, the cumulative Jamming-to-Signal ratio of each pulse train rises with increased interactions with the radar. This indicates that these three algorithms empower the jammer to learn the characteristics of frequency-agile pulse signals transmitted by the radar through ongoing interaction, enhancing the jamming effect. This demonstrates the effectiveness of employing reinforcement learning methods for intelligent jamming decision making. Clearly, the GA-Dueling DQN algorithm proposed in this paper converges to the highest total reward more rapidly. Hence, compared to traditional Q learning, DQN, and the original dueling network, this approach enhances the learning efficiency and effectiveness of intelligent decision making by the jammer.

Figure 9 illustrates the significant advantages of deep reinforcement learning methods over Q-learning. Particularly in environments with extensive action spaces, Q-learning fails to explore the optimal solution for each state, leading to convergence towards lower values. In contrast, the DQN algorithm employs a neural network to approximate the Q-value function, allowing for it to learn more complex state-action relationships. However, in the presence of randomly changing radar pulse signals, accumulating experience for each state becomes challenging, and the DQN algorithm struggles to accurately learn the Q-values of all actions, converging only to local optimal solutions. Dueling DQN decomposes the Q-value function into a state value function and an action advantage function, enabling the model to more accurately estimate the value of a state and the advantage of each action. Additionally, this stable state value estimation reduces variance during learning, enhancing learning efficiency and stability. Therefore, the reward curve of the Dueling DQN method is observed to be more stable than that of the DQN method. However, in the case of randomly occurring radar pulse signals, Dueling DQN still encounters the challenge of learning the optimal action for each state.

The GA-Dueling DQN method proposed in this paper addresses the shortcomings of DQN and Dueling DQN by incorporating GRU layers and attention mechanisms. Incorporating GRU layers enables the model to retain state information across multiple time steps, thereby enhancing model stability against randomness and uncertainty in the environment. The attention mechanism assists the model in automatically screening information more relevant to the current state, thereby reducing interference from irrelevant information. Additionally, benefiting from Dueling DQN’s decomposition of the Q-value function into a state value function and an action advantage function, enhancing the understanding of state values and action importance, the Q-value of each action is estimated more precisely. In random state environments, employing the aforementioned methods enables the model to converge to the optimal solution more rapidly.

### 4.2. Intelligent Frequency Selection by Jammer under Reconnaissance Guidance

Furthermore, the experiment compared the case where the jammer can obtain some prior information about the radar. It is assumed that the cognitive jammer can obtain the carrier frequency of the first subpulse of each radar pulse through interception and parameter estimation. Also, the cognitive jammer knows that although frequency agility is performed between radar subpulses, the change is within the subband range. Jamming decisions are made on this basis. Q learning, DQN and Dueling DQN as well as the GA-Dueling DQN algorithm proposed in this paper are compared with and without reconnaissance information, and the results are shown in Figure 10.

The jamming success probability of the last pulse train in the simulation experiment is shown in Table 3.

Figure 10 illustrates that when utilizing the scouted subpulse frequency as prior knowledge, these four methods can all achieve improved jamming rewards. This is because the acquisition of prior knowledge greatly reduces the jamming decision space. The experimental results show that after using prior knowledge, the jamming success rates of the original Dueling DQN, DQN and Q learning methods significantly improved. Among them, Q learning increased from 45.32% to 75.01%, the DQN method from 72.98% to 82.56%, and the original Dueling DQN method from 68.66% to 89.27%. The improved method proposed in this paper can achieve an over 97% jamming pulse hit rate without radar prior knowledge. With radar prior knowledge, the jamming hit rate of the proposed method is 99.41%, with a difference of less than 3%. Therefore, the method proposed in this paper has significant superiority over the other three methods, and can achieve very high jamming success rates without prior knowledge of radar subpulses.

Figure 11 shows the carrier frequencies of the first 100 main pulses of the radar and the jammer in the last episode. It can be seen that the carrier frequencies of the jammer and radar are almost identical. In fact, in the 10,000 pulses of the last episode, over 97% of the jamming pulses have subpulses that are all identical to the radar subpulses. This shows that under the condition that the frequency agile radar randomly switches carrier frequency, the GA-Dueling DQN method proposed in this paper can successfully jam each subpulse of the radar with a probability of 97.14%.

## 5. Conclusions

This paper models the interaction between the cognitive jammer and the intra-pulse frequency agile radar as a Markov decision process. Based on this model, a deep reinforcement learning algorithm is used to study the optimal jamming frequency selection problem. The jammer can sense radar frequency changes in the environment, adaptively change the jamming frequency, and reduce the radar signal-to-noise ratio, thereby protecting the target from detection. The GA-Dueling DQN method proposed in this paper introduces GRU layers and attention mechanisms to enhance model stability against environmental randomness and estimate the Q-value of each action more accurately. Experimental results demonstrate that the algorithm proposed in this paper achieves significant improvements in Jamming-to-Signal ratio compared to original Dueling DQN, traditional DQN, and Q-learning. Furthermore, for the case with subpulse reconnaissance prior knowledge, the algorithm proposed in this paper holds a distinct advantage in enhancing jamming effects.

For the case with subpulse reconnaissance prior knowledge, Dueling DQN, traditional DQN, Q-learning and the GA-Dueling DQN proposed in this paper all achieved improved jamming rewards.

Comparing radar frequency and jammer frequency allows us calculation of jamming success probability. The improved method proposed in this paper achieves a 97.14% jamming pulse hit rate, and with prior knowledge, the hit rate increases to 99.41%, showcasing significant advantages over the other three methods. This research provides an effective solution to the optimal jamming frequency selection problem and demonstrates the potential of deep reinforcement learning-based jamming strategies to achieve high jamming success rates without relying on fine radar prior knowledge.

## Figures and Tables

**Figure 1 sensors-24-01325-f001:**
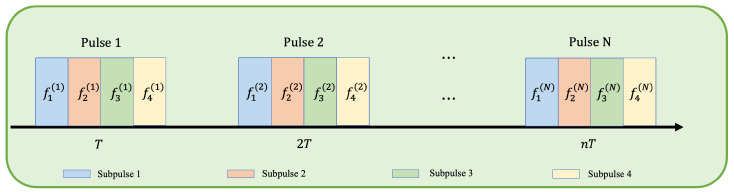
Schematic diagram of the pulse train of a Intra-Pulse frequency agile radar.

**Figure 2 sensors-24-01325-f002:**
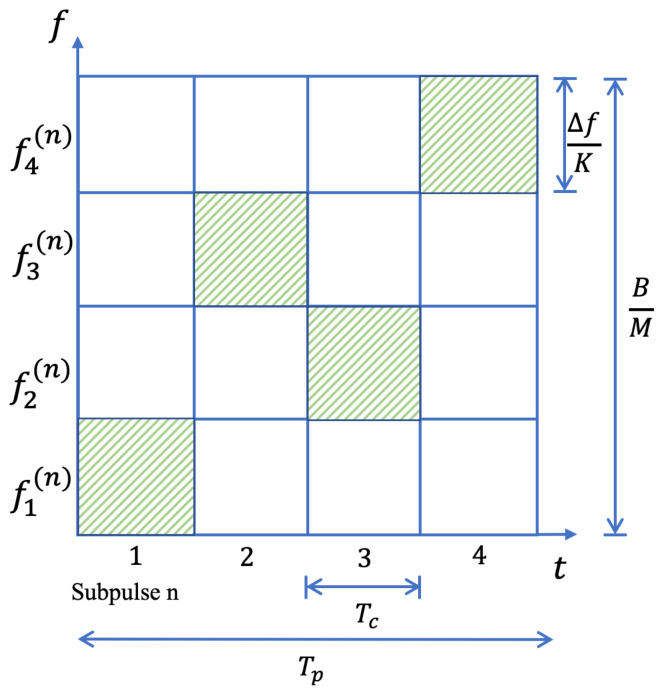
Frequency selection scheme of the intra-pulse frequency agile radar.

**Figure 3 sensors-24-01325-f003:**
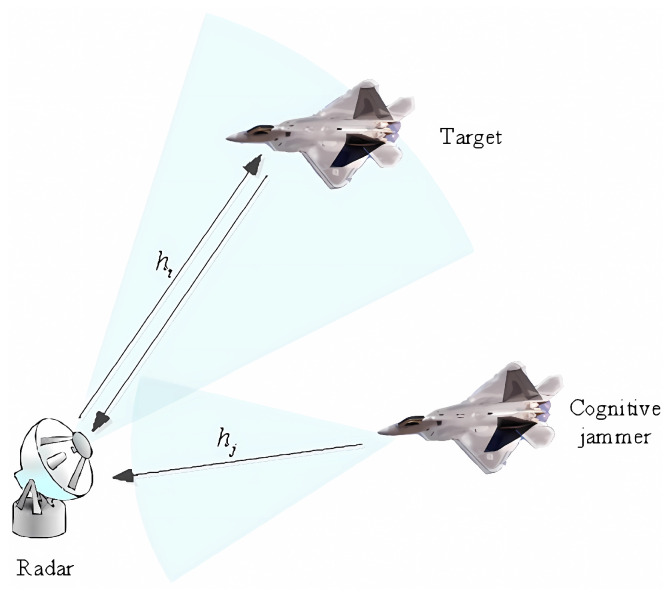
Jammer and radar counter-scenarios.

**Figure 4 sensors-24-01325-f004:**
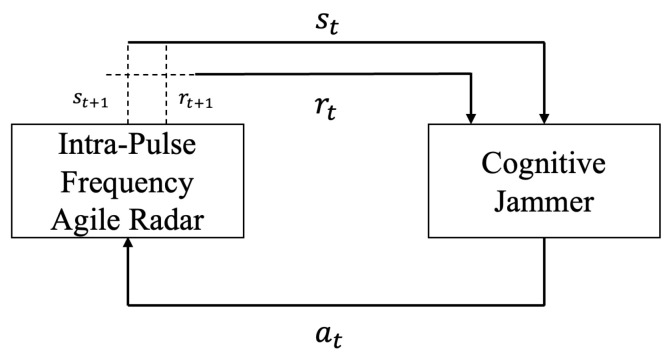
Interaction between cognitive jammer and intra-pulse frequency agile radar.

**Figure 5 sensors-24-01325-f005:**
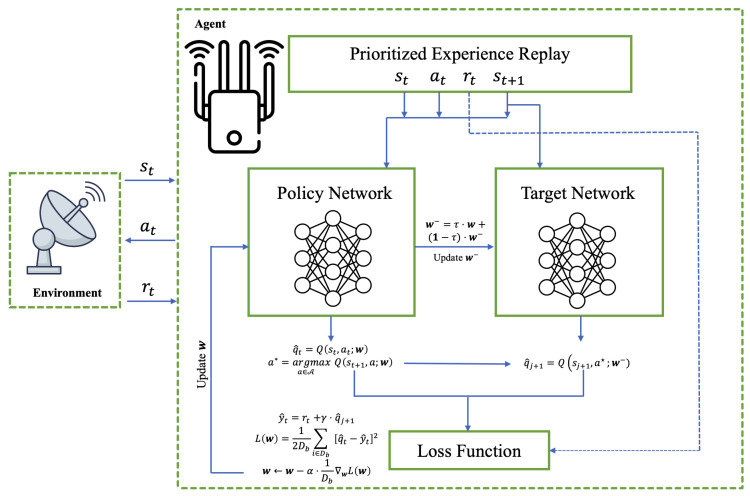
Intelligent jamming strategy algorithm framework.

**Figure 6 sensors-24-01325-f006:**
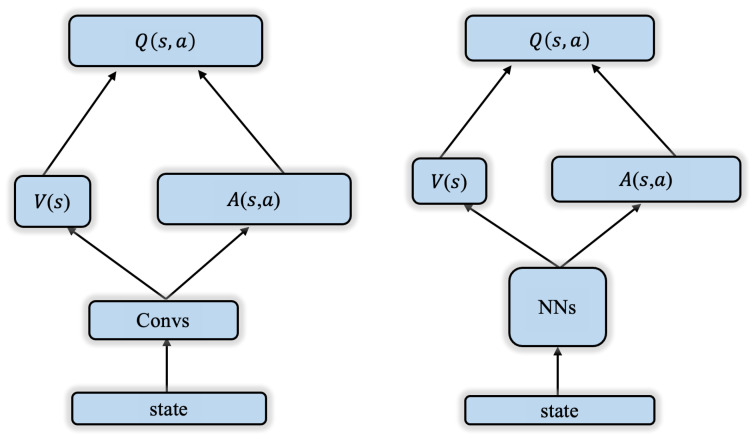
Dueling DQN network structure diagram (**Left**), GA-Dueling DQN network structure diagram (**Right**).

**Figure 7 sensors-24-01325-f007:**
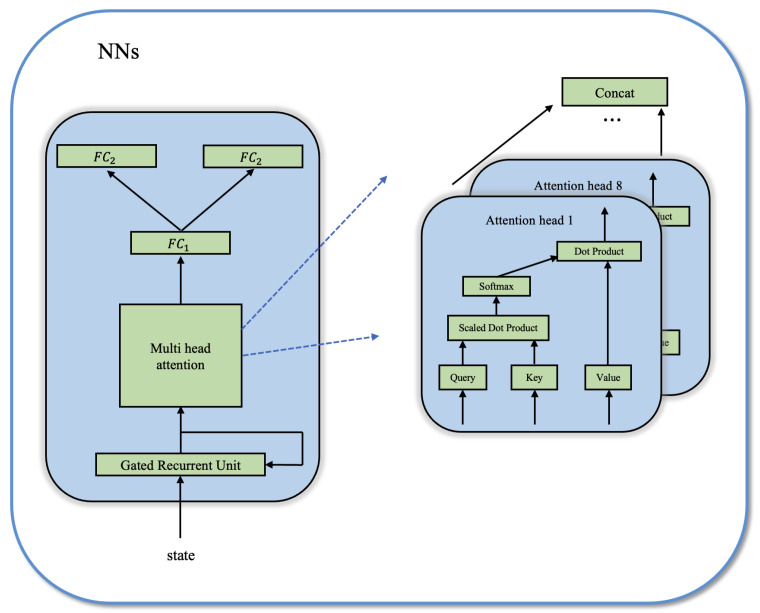
NN structure diagram.

**Figure 8 sensors-24-01325-f008:**
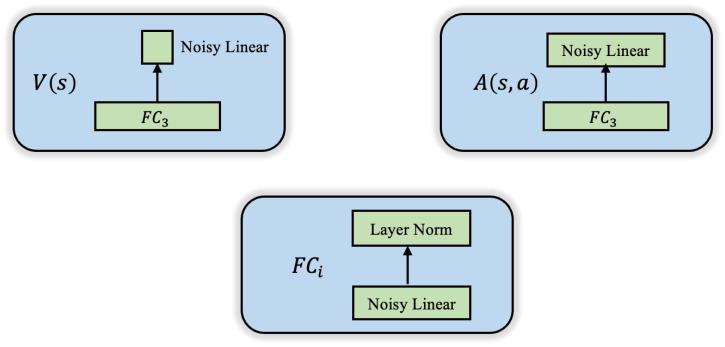
State value network and advantage network structure diagram.

**Figure 9 sensors-24-01325-f009:**
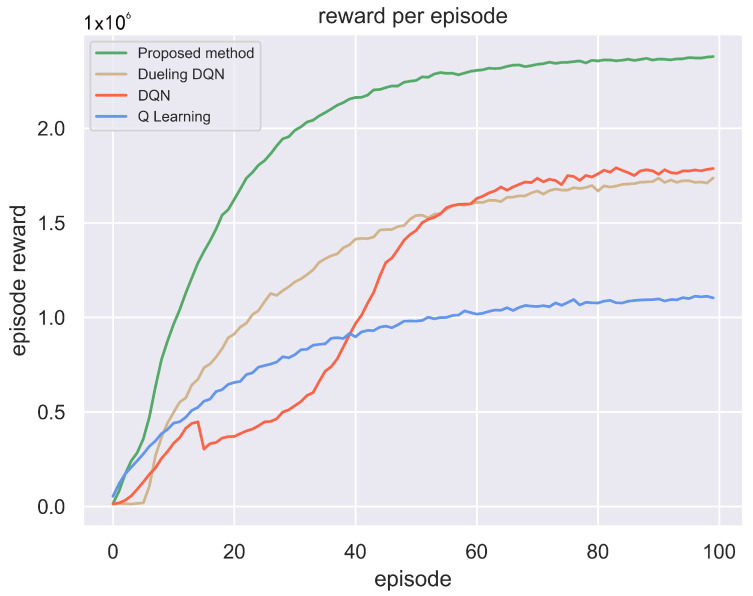
Comparison of Total Rewards for the Jammer in Each Round of the Adversarial Process.

**Figure 10 sensors-24-01325-f010:**
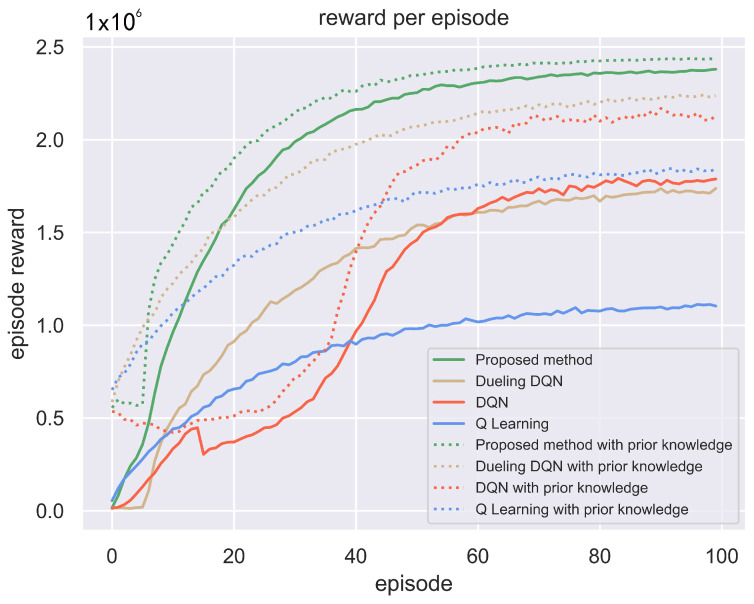
Comparison of jammer gains with and without sub-pulse scouting a priori information case.

**Figure 11 sensors-24-01325-f011:**
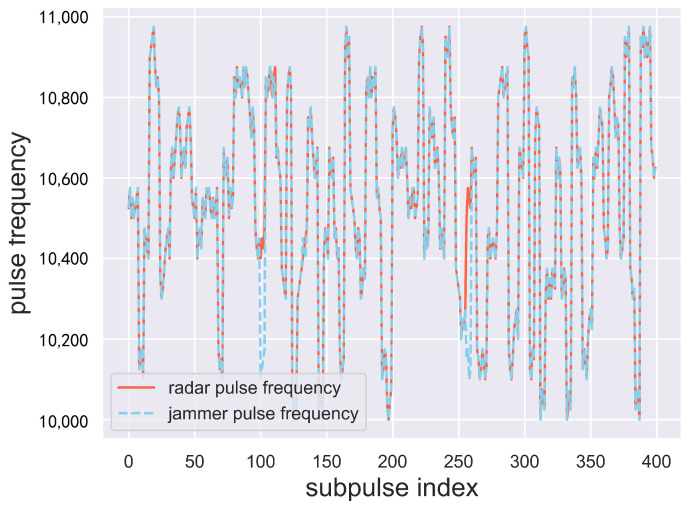
Comparison of sub-pulse carrier frequency of jammer and radar.

**Table 1 sensors-24-01325-t001:** Radar and jammer parameters.

Parameter	Notations	Value
Radar pulse amplitude	$A_{R}$	1 V
Jammer pulse amplitude	$A_{J}$	5 V
Radar-target channel gain	$h_{t}$	0.1 dB
Radar-jammer channel gain	$h_{j}$	0.1 dB
Target RCS	σ	0.1 m^2^
Sidelobe loss	*L*	0.05 dB

**Table 2 sensors-24-01325-t002:** Parameter settings of Networks.

Parameter	Value
GRU	(240, 128)
MultiHeadAttention	(128, 8)
${FC}_{1}$	(128, 64)
${FC}_{2}$	(64, 64)
${FC}_{3}$	(64, 64)
γ	0.9
epsilon start	0.995
epsilon end	0.005
learning rate	0.009
batchsize	256

**Table 3 sensors-24-01325-t003:** Jamming success probability.

Algorithm	Jamming Success Probability
GA-Dueling DQN	**0.9714**
Dueling DQN	0.6866
DQN	0.7298
Q-Learning	0.4532
GA-Dueling DQN with prior knowledge	**0.9941**
Dueling DQN with prior knowledge	0.8927
DQN with prior knowledge	0.8656
Q-Learning with prior knowledge	0.7501

## Data Availability

Data are contained within the article.

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
