# Peer review of "GA-Dueling DQN Jamming Decision-Making Method for Intra-Pulse Frequency Agile Radar"

_sensors, 2024, doi:10.3390/s24041325_

Round 1

Reviewer 1 Report

Comments and Suggestions for Authors

Before final consideration for eventual publication, authors should improve some important aspects of the paper:

- Abbreviations should be defined when used for the first time (see "DQN" in Abstract)

- Technical preparation and typing errors should be improved (missing dot in line 27, "an jamming" and sentence structures in lines 63-66, "poses" in line 103,  repeating the same sentence twice within the first paragraph of section 2.1, "PJ and PR respectively", etc.)

- All variables should be defined (like Tc in eq. (2), Tp in Fig. 2, St after eq. (8), etc.)

-  Figures content should be revised ("nT" in Fig. 1, switching f2(n) and f3(n) in Fig. 2, etc.)

- Confusion with JSR: in eq. (5), page 6, JSR is defined as "jam-to-signal ratio", but directly after it's written that "minimizing" this value is the task of jammer(!) - this is not correct. It's quite opposite: jammer is trying to maximize the effects of jamming. Later, on page 8, authors claim that jammer is trying to "maximize signal-to-jamming ratio.. as a reward". Again, this is quite opposite. But at the bottom of the same page, authors say that reward function rt is signal-to-jamming ratio, which equals JSR according to eq. (15). This is now in obvious collision with content from page 6. But indeed the jammer should maximize rt=JSR, if that was the point. Still, this "reward" is not "received by the radar" (line 308) - it's received by jammer actually?

- It remains unclear which terms in eq. (15) are known by the jammer? Which terms are being predicted / interpolated by the jammer? How rt is calculated finally, and reported values in experiments are achieved?

- Connection between Fig. 5, Fig. 6 and Fig. 7 is unclear. Content of Fig. 5 should be revised and explained properly. Where is V and A in structure presented there? FC1, FC2 and FC3 are standing for what exactly? Terms in Algorithm 1 should be explained (like epsylon and its "greedy strategy", etc.).

- Is there any rule for frequency hopping at sub-pulse level within the radar? Or this is completely random? This is not clear, nowhere explained, nor can be concluded from Fig. 11. Also, how exactly the jammer makes some excellent predictions, with having some wrong conclusions at just a few positions and why - this is also unclear.

- Finally, in ref. [25] "Proceedings of the" is repeated twice.  

Comments on the Quality of English Language

Should be carefully checked.

Reviewer 2 Report

Comments and Suggestions for Authors

1. There are some grammatical errors in the article. For example, in the first line of the third page, two words "modeled" appear; in the second paragraph of Section 2.2, the description of the transmission power of the radar and jammer is written backwards. Please check carefully to avoid similar mistakes.

2. The research point of the article is very new and innovative. The modeling process is also relatively reasonable. However, as a reinforcement learning model, each round needs to have a termination condition during the convergence process. Can the termination conditions of each round be given in detail?

3. The biggest innovation in the article is based on the improvement of the Dueling DQN algorithm. However, the description of the improved algorithm is not detailed enough. Please give more description of the new algorithm.

4. There is a problem with typesetting. After "the following...:" appears on line 484, the table does not appear and goes directly to the Conclusion. In this case, "...is shown in Table 3" should be written directly.

5. The format of tables in the article needs to be unified. If you want to use a three-part form, use a consistent format across all forms.

6. The reward function is the most critical part of the modeling process. Please give more explanation about reward function.

Comments on the Quality of English Language

some typos should be revised.

Round 2

Reviewer 1 Report

Comments and Suggestions for Authors

Seems like the authors did invest efforts to answer on previously given comments, thank you. I have no further (major) concerns. 

Comments on the Quality of English Language

Mostly fine.